# Knee Extensor Muscle Strength Is More Important Than Postural Balance for Stair-Climbing Ability in Elderly Patients with Severe Knee Osteoarthritis

**DOI:** 10.3390/ijerph18073637

**Published:** 2021-03-31

**Authors:** José Casaña, Joaquín Calatayud, Antonio Silvestre, José Sánchez-Frutos, Lars L. Andersen, Markus D. Jakobsen, Yasmín Ezzatvar, Yasser Alakhdar

**Affiliations:** 1Exercise Intervention for Health Research Group (EXINH-RG), Department of Physiotherapy, University of Valencia, 46010 Valencia, Spain; jose.casana@uv.es (J.C.); yasmin.ezzatvar@uv.es (Y.E.); 2Department of Physical Therapy, University of Valencia, 46010 Valencia, Spain; jose.sanchez-frutos@uv.es (J.S.-F.); yasser.alakhdar@uv.es (Y.A.); 3National Research Centre for the Working Environment, 2100 Copenhagen, Denmark; LLA@nfa.dk (L.L.A.); mdj@nfa.dk (M.D.J.); 4Department of Orthopaedic Surgery, Clinic Hospital of Valencia, 46010 Valencia, Spain; antonio.silvestre@uv.es; 5Department of Orthopaedic Surgery, School of Medicine, Valencia University, 46010 Valencia, Spain; 6Sport Sciences, Department of Health Science and Technology, Aalborg University, 9220 Aalborg, Denmark

**Keywords:** falls, aging, posturography, total knee arthroplasty, postural balance

## Abstract

Knee osteoarthritis is a chronic joint disease which damages articular cartilage. In its severe stages, it results in impairments in balance and muscle strength loss, which affect daily life activities such as walking or climbing stairs. This study sought to investigate associated factors with stair-climbing ability in this population, with special interest in measuring the relevance of postural balance for this task. Forty-four patients scheduled to undergo unilateral total knee arthroplasty were assessed. Timed up and go test, stair ascent–descent test, three different isometric strength tests (knee flexion, knee extension and hip abduction), active knee extension and flexion range of movement and static postural balance assessment were evaluated. Spearman’s correlation coefficients and multiple linear regression analysis determined the strength of association between the different variables and stair-climbing time. No significant association between the stair-climbing time and static balance was found. Significant associations were found between stair-climbing time and timed up and go (r = 0.71; *p* < 0.0001) and maximal knee extensor strength (r = –0.52; *p* = 0.0003). One-year increase in age was associated with 0.15 s (95% CI 0.00 to 0.30) slower stair-climbing time. In conclusion, muscle strength is more important than postural balance for stair-climbing ability in this population.

## 1. Introduction

Knee osteoarthritis (OA) is a chronic joint disease and a leading cause of disability among adults that produces a degradation of articular cartilage, sclerosis of subchondral bone, osteophyte formation on radiographs, joint pain and stiffness [1]. Knee OA commonly results in difficulty in participating in daily life activities [2], which can make work difficult when not impossible, especially for those with physical activities imposing a high mechanical stress at the knee [3]. For instance, during stair climbing, the pressure on the knees increases to six times that of the body weight [4], which in turn is considered a risk factor for knee OA due to occupational activities [3]. Given the increased knee OA prevalence with population ageing and the increasing state pension age in most European, the implications of knee OA should be taken seriously since a lot of older workers will suffer from this.

One of the most relevant impairments in patients with knee OA is the reduced postural balance when compared with age-matched healthy pairs [5,6]. In fact, those patients with a severe stage of the disease have demonstrated more deficits in postural balance than their mild pairs [7]. Together with muscle weakness, poor postural balance is considered the major independent contributor to falls [8]. Patients with knee OA have demonstrated a higher risk of falls, with a study reporting at least one fall in the previous 12 months for 48% of the surgical group in comparison with 30% of the control group [9]. Falling during stair walking may have fatal consequences, being the primary cause of accidental deaths in the elderly [10]. However, the number of non-fatal stair-walking injuries is also high and commonly results in hospitalization and multiple injuries such as femoral neck fractures or other fractures in the leg, arms, neck and trunk [11]. In fact, the frequency and severity of falls have been recently associated with time to health-related work limitation and time to labor-force exit among workers aged 65 years and older [12]. Since stair walking represents a relevant factor for falling [13], the ability to ascend and descend stairs has been used as an indicator of disability and loss of independence and mobility [10]. A poorer mobility measured by the ability to climb stairs at the end stage of OA is a predictor of undergoing a total knee arthroplasty (TKA) [14] and is associated with greater length of stay at the hospital after receiving surgery [15].

Despite the association between lower-limb muscle strength and stair-climbing performance among patients with early symptomatic OA [16] and among patients after receiving a TKA [17,18] having been more investigated, there are no studies evaluating whether postural balance is linked with stair-climbing performance in patients with severe knee OA. Postural balance is commonly assessed by posturography, which is considered the gold-standard measurement for this purpose [2]. Whitchelo et al. [19] highlighted the need to understand the reasons for the difficulties ascending and descending stairs in OA patients. Hence, posturographic measures identifying impairments that are closely linked with stair-climbing performance may provide greater insight about which tests must be selected or prioritized in pre/postoperative assessments and may improve the decision-making for pre/postoperative training programs’ design. The postural balance evaluation can be useful to prevent falls in patients with knee OA, especially at the severe stage of the disease [7]. This study investigated which factors are associated with stair-climbing ability in elderly patients with severe knee osteoarthritis. We were especially interested in evaluating whether postural balance assessed by posturography is associated with this daily living task. It was hypothesized that postural balance would show an association with the ability to ascend and descend stairs.

## 2. Materials and Methods

### 2.1. Patients

A convenient sample of forty-four subjects (7 men, 37 women) with end-stage knee OA participated in the study. Inclusion criteria were as follows: patients above 60 years old with knee OA diagnosed by a surgeon according to the American College of Rheumatology classification [20] and scheduled for TKA in a local hospital during 2014. Participants were excluded if they presented pain in the contralateral limb (maximum pain, ≥4 out of 10 during daily activities) [18], had previously undergone knee joint replacement surgery of the affected joint or any other lower limb surgery within the past year, had any medical condition which contraindicated exercise or if they had any disease that may affect functional performance. Before starting the present study, each participant read and signed an informed consent form. This study was previously approved by the institution’s review boards, and all procedures described in this section comply with the requirements listed in the 1975 Declaration of Helsinki and its amendment in 2008.

### 2.2. Procedures

Each subject participated in 2 sessions: a familiarization and data assessment session. The familiarization session occurred 48–72 h before the first data collection. Participants were required to avoid any stimulants or intense physical activity (i.e., more than normal or regular daily life activity) 12 h before the sessions. In the first session, the subjects were familiarized with all measures that would be taken in the subsequent session. In the first session, the subjects were familiarized with the testing activities that would be performed in the assessment session. Firstly, height and body mass was measured (Tanita model BF- 350, Arlington Heights, IL, USA) and then subjects practiced the exercises typically 1–3 times each until the subject felt confident and the researcher was satisfied that proper execution was achieved. The data assessment session consisted of two functional tasks (timed up and go test and stair ascent–descent test), evaluation of 3 different isometric strength tests (knee flexion, knee extension and hip abduction), active knee extension and flexion range of motion (ROM) and static postural balance assessment (Romberg eyes open and Romberg eyes closed). All the measurements were performed at the Physical Therapy Faculty by the same rater (a professor with previous experience in these measurements).

Static postural balance. For the measurement of the static postural balance, a NEDSVE/IBV force platform was used (Biomechanics Institute of Valencia, Valencia, Spain). Two tests with different levels of difficulty were performed: Romberg test with eyes open (REO) and Romberg test with eyes closed (REC). For the REO, participants were asked to remove their shoes and stand motionless with both feet drawing a 30° angle, trying to coincide with the feet contour drawn on the platform, and maintaining their balance during all the test with eyes open, starting at a single point marked by the therapist. The patients were instructed have their arms hanging down the side and to stand as still as possible. The performance of the REC was similar to REO, but eyes remained closed during the entire test, in order to avoid use of vision.

Each test was performed three times during 30 s and the mean average balance values were calculated for its analysis. The sampling rate of the system was set at 40 Hz according to the manufacturer’s recommendations [21]. A trial was discarded and repeated if participants removed the feet from the platform before finishing the test or were completely unable to maintain balance. The NEDSVE/IBV dynamometric platform proved to be a reliable and valid instrument for the functional assessment of balance disorders [21].

Stair-climbing test. Subjects were asked to ascend and descend a flight of 4 stairs once (each step was 50 cm wide, 15 cm high and 25 cm deep) as quickly but safely as possible. Patients were asked to stand at the bottom of the first step, to go up the stairs, turn around on the top step and come all the way down until both feet were on the floor. They were instructed to use the handrails as needed while ascending or descending the stairs. Total time taken to complete this task was measured in seconds with a stopwatch which was stopped once the patient reached the start line after ascent and descent of the stairs. The test was performed 2 times with a 30-s rest period between repetitions, and the average time to complete the task was recorded [18]. A test–retest reliability coefficient of 0.93 has been reported for this assessment [22].

Timed up and go. The timed up and go test (TUG) is a simple test to assess mobility, lower extremity function and fall risk which does not require any specific equipment. The TUG measures the time that a person takes to rise from a standard arm chair (not using their arms to stand up), walk to a line on the floor 3 m away, turn around, walk back to the chair and sit down again [23]. The outcome of the test was the time to complete the task, with shorter times indicating better performance [23]. Subjects were permitted to use walking aids if necessary, and the time was measured in seconds with a chronometer. The TUG test is highly reliable when performed in elderly populations [24].

Isometric strength with dynamometer. For measuring the isometric strength, a portable hand-held dynamometer was used (Nicholas Manual Muscle Tester, Lafayette Instruments, Indiana, USA). The maximal isometric knee flexion and extension strength tests were performed according to a previously described technique [25]. Patients were seated in the edge of the plinth, with their thighs in contact with the examination table and with the hip drawing a constant angle (90°), not leaning their trunk backward. They were instructed to produce as much force as possible against the dynamometer, held by the examiner, who avoided overcoming the subject’s effort. One practice trial was given before the assessment. For measuring the isometric knee extension strength, the dynamometer was positioned perpendicular to the tibia, proximal to the ankle, and fixated by a belt to the plinth. For measuring the isometric knee flexion strength, it was placed on the posterior aspect of the lower leg, anchored by a belt to the handlebar of a glass suction cup on the wall. For measuring the isometric hip abduction strength, a previously described technique was used [26]. The patient was placed supine, with one leg extended over the examination table and the other leg flexed. Dynamometer was fixed with a belt towards the wall and held by the examiner, and the patient was instructed to exert an abduction maximal contraction with the flexed leg. All patients performed three 5-s maximal voluntary isometric contractions per each measurement, with a build-up phase of 2 s and 3 s where steady maximal force exertion was performed. Standardized encouragement (i.e., “go, go, go”) for maximal force exertion was provided by the rater. Mean maximal strength of the three repetitions was calculated for its analysis. Hand-held dynamometer had good inter- and intrarater reliability values for knee flexors (ICC range 0.76–0.94) and excellent values for knee extensors (ICC range 0.92–0.97) in inpatients awaiting TKA in both the affected and unaffected knee [25]. In addition, the aforementioned isometric abduction strength test has shown excellent reliability with an ICC of 0.85 [26].

Active knee ROM. A digital goniometer (Digital Absolute Axis Goniometer, Baseline Evaluation, White Plains, NY, USA) was used for measuring active knee ROM. All goniometric measurements were performed according to the technique described in [27]. The subject was placed supine for the measurement of the active knee flexion and extension ROM, with extended knees and the hip in a neutral position and with the upper thigh exposed so that the greater trochanter could be visualized. Subjects were asked to actively flex and extend their knee as far as possible to the maximum. A towel was placed under the ankle to allow for maximum knee extension. For standardization purposes and guarantee of repeatability, each subjects’ bony landmarks were identified and marked by the main investigator with a dermographic marker pen. The goniometer was positioned with its center fulcrum over the lateral epicondyle of the femur, the proximal arm was aligned with the lateral midline of the femur, using the greater trochanter for reference, and the distal arm was aligned with the lateral midline of the fibula, using the lateral malleolus and fibular head for reference. Measurements were made three times, taking the average value to analyze data. Assessment of the knee ROM in patients with knee OA showed high reliability values, with a coefficient of 0.96 for flexion and 0.81 for extension [28].

### 2.3. Statistical Analysis

All statistical analyses were performed using the SAS statistical software for Windows (SAS Institute, Cary, NC, USA). Spearman’s correlation coefficients were used to assess the associations between the different outcomes. In addition, we performed a multiple linear regression analysis with backward elimination (*p* < 0.10 to stay in the model), where stair time was the dependent variable and age, gender, BMI, knee flexor strength, knee extensor strength, knee flexion ROM, knee extension ROM and postural balance were the independent variables.

## 3. Results

The mean age of participants was 66.7 years (range 60–75 years). Table 1 shows the complete mean values for the measured outcomes. The Spearman correlations revealed no correlation between the stair-climbing test and the static postural balance tests as well as with knee ROM measures. Significant correlation values were found between the stair ascent and descent test with the TUG (r = 0.71, *p* < 0.01) and the isometric knee extension test (r = –0.52, *p* ≤ 0.01). Table 2 shows the complete data for the associations between the different variables. Results from the multiple linear regression with backward elimination showed that age and knee extensor strength were significantly associated with stair-climbing time, with standardized beta values of 0.27 (*p* = 0.05) and −0.46 (*p* < 0.01), respectively. Each year increase in age was associated with a 0.15 s (95% CI 0.00 to 0.30) increase in stair-climbing time, and each kg increase in knee extensor strength was associated with a 0.14 s (95% CI −0.23 to −0.05) decrease in stair-climbing time.

## 4. Discussion

The main finding in our study was that static postural balance assessed by posturography was not a major determinant of the ability to climb stairs in patients with end-stage OA. We found that this ability declines with age but improves with knee extensor muscle strength.

In contrast of our hypothesis, postural balance measured by posturography showed an absence of significant associations with the stair-climbing test. There are no previous studies evaluating the association between these tests and stair-climbing capacity in end-stage OA patients. A previous study [29] found that in 76-year-old participants, the measures of functional (i.e., walking in a figure of 8) and static postural balance (i.e., one-leg stance) had a moderate significant association with the capacity to climb a step. More recently, the ability to climb high steps was significantly correlated with a static balance test, which consisted of measuring the time in a one-leg stance and the ability to perform at least 30 s of tandem stance, in elderly women [30]. However, the aforementioned studies were not conducted in an OA population and used different static balance tests as well as different climbing tests (measured by the ability to mount boxes of increasing heights without support), compared to the current study. A relationship between static postural balance and stair-climbing ability can be indirectly hypothesized after the results of an intervention study. For example, a recent study found that the double-limb and single-limb static postural control was not improved after an 8-week stair-climbing intervention in senior participants, whereas the dynamic balance measures improved [31]. Proper stair climbing involves the correct production strength but also requires the maintenance of a constant postural balance, since the body’s center of mass and the base of support are constantly changing while the person has to adapt movement to the stair environment or their own characteristics [32]. Hence, it seems that stair-climbing performance is especially influenced by more dynamic rather than static balance tasks. In agreement with this notion, the TUG test showed the highest association with the stair-climbing performance, as was hypothesized. This fits in with previous results reporting that TUG was significantly linked to the ability to mount boxes of increasing heights [30]. The sit-to-stand component of the TUG requires leg force production, movement and acceleration of the center of mass and postural stabilization [33], similarly to the tasks and abilities required for stair climbing that were previously explained. Furthermore, stepping, acceleration and deceleration required during the TUG [23] are also present during the stair-climbing test.

Knee strength was moderately associated with the stair-climbing test. As proper stair climbing requires the effective production of primarily concentric muscle forces to ascend them [32], the correlation between those tests seems logical. Associations between leg extension strength and functional impairments and altered movement patterns in knee OA patients have been reported [34]. In fact, a recent study found that extension/flexion muscle strength decreases were linked with walking and stair-climbing impairments over a 3-year period in early symptomatic OA patients [16]. Even 1 year [17] or 2 years [18] after receiving a TKA, greater preoperative quadricep strength was one of the strongest predictors of stair-climbing ability. In addition, knee extensor and flexor muscle power deficits [35] and rate of force development [36] have been found to influence stair-climbing and walking variables, respectively. All these factors together lead to a distortion of the correct movement patterns. For instance, less knee flexion during walking is used by OA patients with weak quadriceps, longer muscle activity times and greater co-contractions than gender-matched control subjects [37].

Knee ROM was not related with the ability to ascend and descend stairs in end-stage OA patients. Mizner et al. [17] reported no relationship between preoperative knee ROM and stair climbing and TUG performance one year after TKA. In the same vein, Zeni and Snyder-Mackler [18] found that preoperative active knee ROM was not a predictor of handrail use during stair ascent and descent two years after undergoing a TKA. Altogether, these data suggest that preoperative knee ROM does not predict stair-climbing ability neither before nor after surgery, so it should not be considered a key aspect of preoperative training programs aiming to improve this functional task.

Regarding the association between age and stair-climbing time, our results support previous findings showing that age increases were related to handrail use and worse stair-climbing performance [18]. Age-related reductions in muscle mass and strength have been well described during the past years, being directly linked to functional impairments and disability [38]. It needs to be considered that the relationship between other factors such as comorbidity (e.g., hypertension or type 2 diabetes) and stair-climbing ability were not evaluated, and this remains a limitation [39]. In addition, joint status (e.g., meniscus, cartilage and bone status) might also play a role [40].

## 5. Conclusions

Static postural balance measured by posturography and knee ROM are not related with stair-climbing ability in patients with end-stage knee OA. On the contrary, those with lower isometric knee extensor strength and reduced TUG ability would exhibit an impaired ability to ascend and descend stairs during the end-stage of the disease. These findings highlight the importance of assessing these variables to evaluate mobility impairment and the ability to perform normal daily activities in this population. Additionally, our results suggest that strength training and the combination of tasks involving strength, walking and dynamic postural balance must be prioritized in training programs before undergoing TKA when improvements in stair efficacy are desired, while static postural balance and knee ROM training would provide less positive results.

## Figures and Tables

**Table 1 ijerph-18-03637-t001:** Mean values.

Variable	Mean	SD	Min	Max
Age (years)	66.7	4.0	60	75
Weight (kg)	81.5	10.7	65.6	104
Height (meters)	1.6	0.1	1.45	1.78
BMI (kg/m^2^)	31.8	3.9	26.7	45.0
Timed up and go (seconds)	8.7	1.7	6.9	15.8
Stair-climbing test (seconds)	11.2	2.3	7.3	19.7
Isometric knee flexion (kg)	9.3	3.7	5.2	20.9
Isometric knee extension (kg)	21.7	7.1	12.7	38.1
Isometric hip abduction (kg)	7.2	1.3	4.5	9.56
Knee flexion range of motion (degrees)	104.2	17.1	79	130
Knee extension range of motion (degrees)	14.6	6.1	2	27
Romberg eyes open total sway area (mm^2^)	45.1	6.3	29.9	56.44
Romberg eyes open medial–lateral displacement (mm)	14.9	1.8	12.1	19.99
Romberg eyes open antero-posterior displacement (mm)	18.3	2.1	13.1	23.5
Romberg eyes open velocity (m/s)	0.01	0	0.01	0.02
Romberg eyes closed total sway area (mm^2^)	94.3	11.8	68.7	117.6
Romberg eyes closed medial–lateral displacement (mm)	24.9	2.4	20.7	31.5
Romberg eyes closed antero-posterior displacement (mm)	26.7	2.2	23.2	31.8
Romberg eyes closed velocity (m/s)	0.02	0	0.01	0.03

**Table 2 ijerph-18-03637-t002:** Correlation coefficients between the stair ascent and descent test and the other variables.

Variables	*r* ^a^	*p* Value
Age	0.10	0.5098
Weight	−0.10	0.5164
Isometric knee flexion	−0.15	0.3315
Isometric knee extension	−0.52	0.0003 *
Isometric hip abduction	0.01	0.9316
Timed up and go test	0.71	<0.0001 *
Knee flexion range of motion	−0.14	0.3498
Knee extension range of motion	−0.21	0.1617
Romberg eyes open (REO):		
REO total sway area	0.07	0.651
REO antero-posterior displacement	−0.23	0.1267
REO medial–lateral displacement	−0.18	0.2424
REO velocity	−0.11	0.4935
Romberg eyes closed (REC):		
REC total sway area	−0.06	0.6935
REC antero-posterior displacement	0.02	0.8929
REC medial–lateral displacement	−0.01	0.9521
REC velocity	−0.17	0.2821

^a^ Spearman’s correlations. * Denotes significant correlations.

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
