# Peer review of "Knee Extensor Muscle Strength Is More Important Than Postural Balance for Stair-Climbing Ability in Elderly Patients with Severe Knee Osteoarthritis"

_ijerph, 2021, doi:10.3390/ijerph18073637_

Round 1
Reviewer 1 Report
The authors aim to compare muscle strength with posture balance in association with stair climbing ability. it is not surprised to find the muscle strength is more important. A few issue points to address:
- OA is a whole joint disease, not only muscle related to stair climbing, but also meniscus, cartilage and bone pathology might also play a role. More citations on relevant literatures are needed, such as
Wen CY, Lu W, Chiu KY. Importance of subchondral bone in pathogenesis and management of osteoarthritis - from bed to bench. Journal of Orthopaedic Translation 2014 Jan 2(1) 16-25.
2. The authors did not take the comorbidity of knee osteoarthritis into consideration such as diabetes and hypertension. Please add it into the limitation of the present study.
Wen C, Chen Y, Tang HL, Yan CH, Lu WW, Chiu KY. Bone loss at subchondral plate in knee osteoarthritis patients with hypertension and type 2 diabetes mellitus. Osteoarthritis & Cartilage. 2013 Nov; 21(11): 1716-23.
Author Response
Reviewer 1
Comments and Suggestions for Authors
The authors aim to compare muscle strength with posture balance in association with stair climbing ability. it is not surprised to find the muscle strength is more important. A few issue points to address:
- OA is a whole joint disease, not only muscle related to stair climbing, but also meniscus, cartilage and bone pathology might also play a role. More citations on relevant literatures are needed, such as
Wen CY, Lu W, Chiu KY. Importance of subchondral bone in pathogenesis and management of osteoarthritis - from bed to bench. Journal of Orthopaedic Translation 2014 Jan 2(1) 16-25.
- The authors did not take the comorbidity of knee osteoarthritis into consideration such as diabetes and hypertension. Please add it into the limitation of the present study.
Wen C, Chen Y, Tang HL, Yan CH, Lu WW, Chiu KY. Bone loss at subchondral plate in knee osteoarthritis patients with hypertension and type 2 diabetes mellitus. Osteoarthritis & Cartilage. 2013 Nov; 21(11): 1716-23.
Our response: we have added this at the discussion/limitations
Reviewer 2 Report
This research in spite of relatively simple design and hypothesis has each part done by very high quality. It makes reading of this article very impressive. The results are completely discussed and each conclusion supported by data of the research.
Author Response
Reviewer 2
Comments and Suggestions for Authors
This research in spite of relatively simple design and hypothesis has each part done by very high quality. It makes reading of this article very impressive. The results are completely discussed and each conclusion supported by data of the research.
Our response: We thank this positive feedback.
Reviewer 3 Report
Thank you for the opportunity to review this manuscript, which considers some interesting, applied issues. This study appears to be novel, but as submitted needs considerable work on the presentation. The authors showed an interesting point about the “Knee extensor muscle strength is more important than postural balance for stair climbing ability in elderly patients with severe knee osteoarthritis”, unfortunately, there are several points to overcome.
Abstract:
I suggest to include the significance levels (p=….)
Materials and methods:
I suggest to include the inclusion/exclusion criteria
For each test, the authors need to include the reference related to the validity of each test
For each device used, the authors need to include more details (i.e., sample frequency (Manufactory, Model/Version, City, Coutry); moreover, they need to include accuracy/precision for each one
Line 165-169: The ICC related on other study is misleading for the readers; because the ICC should be made for this study
Statistical analysis
The normality of distribution analyis is missing; please check…
I suggest to perform Principal Component Analysis (PCA) to find the main component summarizing…….as illustred by Kollias et al. (Kollias I., Hatzitaki V., Papaiakovou G., Giatsis G. (2001). Using principal components analysis to identify individual differences in vertical jump performance. Res. Q. Exerc. Sport. 72 63–67. 10.1080/02701367.2001.10608933)
In order to assess test sensitivity, I suggest resort to weighing smallest worthwhile change (SWC) against SEM, focusing on the thresholds proposed by Liow and Hopkins (Liow D. K., Hopkins W. G. (2003). Velocity specificity of weight training for kayak sprint performance. Med. Sci. Sports Exerc. 35 1232–1237. 10.1249/01.MSS.0000074450.97188.CF)
Author Response
Reviewer 3
Comments and Suggestions for Authors
Thank you for the opportunity to review this manuscript, which considers some interesting, applied issues. This study appears to be novel, but as submitted needs considerable work on the presentation. The authors showed an interesting point about the “Knee extensor muscle strength is more important than postural balance for stair climbing ability in elderly patients with severe knee osteoarthritis”, unfortunately, there are several points to overcome.
Abstract:
I suggest to include the significance levels (p=….)
Our response: done as indicated
Materials and methods:
I suggest to include the inclusion/exclusion criteria
Our response: done as indicated
For each test, the authors need to include the reference related to the validity of each test
For each device used, the authors need to include more details (i.e., sample frequency (Manufactory, Model/Version, City, Coutry); moreover, they need to include accuracy/precision for each one
Our response: this information has been provided
Line 165-169: The ICC related on other study is misleading for the readers; because the ICC should be made for this study
Our response: it is widely accepted that studies can report previous reliability data from other studies as reference, especially if procedures are conducted in accordance, as in our case
Statistical analysis
The normality of distribution analyis is missing; please check…
Our response: For multiple linear regression the residuals have to be normally distributed, and they were.
I suggest to perform Principal Component Analysis (PCA) to find the main component summarizing…….as illustred by Kollias et al. (Kollias I., Hatzitaki V., Papaiakovou G., Giatsis G. (2001). Using principal components analysis to identify individual differences in vertical jump performance. Res. Q. Exerc. Sport. 72 63–67. 10.1080/02701367.2001.10608933)
In order to assess test sensitivity, I suggest resort to weighing smallest worthwhile change (SWC) against SEM, focusing on the thresholds proposed by Liow and Hopkins (Liow D. K., Hopkins W. G. (2003). Velocity specificity of weight training for kayak sprint performance. Med. Sci. Sports Exerc. 35 1232–1237. 10.1249/01.MSS.0000074450.97188.CF)
Our response: While these analyses could also be performed they are not necessary for the purpose of the present article. Thus, we have chosen to stick to our pre-planned analyses.
Round 2
Reviewer 1 Report
N/A
Reviewer 3 Report
the main docment wasa well improved